# Organosilicon uptake by biological membranes

Pepijn Beekman[1,2,6], Agustin Enciso-Martinez [3,6], Sidharam P. Pujari [2], Leon W. M. M. Terstappen [3], Han Zuilhof [2,4,5], Séverine Le Gac[1✉] & Cees Otto [3✉]

Organosilicon compounds are ubiquitous in everyday use. Application of some of these compounds in food, cosmetics and pharmaceuticals is widespread on the assumption that these materials are not systemically absorbed. Here the interactions of various organosilicon compounds (simeticone, hexamethyldisilazane and polydimethylsiloxane) with cell membranes and models thereof were characterized with a range of analytical techniques, demonstrating that these compounds were retained in or on the cell membrane. The increasing application of organosilicon compounds as replacement of other plastics calls for a better awareness and understanding of these interactions. Moreover, with many developments in biotechnology relying on organosilicon materials, it becomes important to scrutinize the potential effect that silicone leaching may have on biological systems.

[1] Applied Microfluidics for BioEngineering Research, MESA+ Institute for Nanotechnology & TechMed Center, University of Twente, Enschede, The Netherlands. [2] Laboratory of Organic Chemistry, Wageningen University, Wageningen, The Netherlands. [3] Medical Cell BioPhysics, TechMed Center, University of Twente, Enschede, The Netherlands. [4] School of Pharmaceutical Science and Technology, Tianjin University, Tianjin, China. [5] Department of Chemical and Materials Engineering, Faculty of Engineering, King Abdulaziz University, Jeddah, Saudi Arabia. [6] These authors contributed equally: Pepijn Beekman, Agustin Enciso-Martinez. ✉email: s.legac@utwente.nl; c.otto@utwente.nl

Organosilicon compounds, which are silicon-containing hydrocarbons, have a wide range of accepted usage[1]. Examples include silicones like polydimethylsiloxane (PDMS) which is accepted by the European Food Safety Authority (EFSA) as a food additive[2], and widely used in the field of microfluidics[3]. The silicone-based over-the-counter drug simeticone is used as gastro-intestinal surfactant to treat colic in infants[4]. Interestingly, there is no dosage limitation for this drug since it is claimed not to be absorbed systemically, and it has been generally recognized as safe since before the FDA started Over-The-Counter Drug Review in 1972[5]. However, despite the widespread use of silicones in products for use in humans, there is relatively little literature regarding the possible interactions between silicone molecules and lipid membranes and, potentially, other biomolecules vital to living organisms. Moreover, various research disciplines use silicones for a broad range of applications. Leaching of low-molecular weight components into the samples under study may influence results[3,6,7], for instance in lab-on-a-chip research and scanning electron microscopy (SEM) studies, which we briefly clarify below:

1. In the lab-on-a-chip field[8,9], the massive adoption of PDMS for the production of microfluidic and organ-on-a-chip platforms can be attributed to the ease of device fabrication, its optical transparency, its gas-permeability, which is particularly attractive for cell culture experiments, and its elastomeric properties[10]. Finally, PDMS has proven to be biocompatible in the sense that it does not significantly affect cell viability, also for very sensitive cells like embryos, primary cells and ex vivo ovarian tissues[7,11–13]. In-vivo short term studies reported no significant changes in survival of rats that were fed diets containing up to 10% PDMS[14,15]. However, to the best of our knowledge, systemic uptake was never comprehensively studied. In contrast, it has been demonstrated that PDMS in the cells' microenvironment do modulate gene expression profiles significantly[16], especially in comparison with other polymers[17].

2. In SEM studies, biological samples require pretreatment before they can be placed in a vacuum chamber for imaging. Fixation, dehydration, drying, and coating with an electrically conductive layer are typically required. Dehydration and drying seem to be the most critical steps as they can give rise to artifacts, such as specimen shrinkage and distortion[18–23]. A common method for drying uses hexamethyldisilazane (HMDS)[24,25]. The mechanism proposed by which HMDS interacts with biological specimens has been via transfer of trimethylsilyl groups[26], which can happen with, e.g., sugars and amino acids in biological specimens. In this process proteins crosslink to fix the biological specimen, preventing it from collapsing during drying[25].

Based on preliminary observations of silicone residue in lipid membranes after incubation in microfluidic channels and HMDS-drying (vide infra), we set to investigate whether organosilicons, broadly speaking, interact with biological membranes. To illustrate the generality of the silicone-membrane interactions, three different organosilicon sources were included in this study (See SI1 for structural information):

1. PDMS (Sylgard 184), whose oligomers leach from incompletely cured microfluidic channels.
2. HMDS used for dehydration in electron microscopy sample preparation.
3. Infacol, an over-the-counter drug, containing simeticone, an organosilicon compound similar in structure to PDMS, mixed with silica nanoparticles.

Two different specimens were considered: cells (LNCaP and HT-29 cell lines) and supported lipid bilayers (SLBs) prepared from 1,2-dioleoyl-sn-glycero-3-phosphocholine (DOPC), which is abundant in biological membranes[27]. SLBs are widely used as models of cellular membranes[28]; here, they enable to study organosilicon interactions with phospholipid molecules.

Interactions between silicones and biological membranes were studied using four analytical techniques: a) Confocal Raman micro-spectroscopy, to probe the presence of specific chemical bonds down to 400 nm spatial resolution and map the spectra in a hyperspectral image; b) Auger electron spectroscopy (AES), to identify atomic species present at the surface of a sample (probing depth of 3 nm), and which allows overlay with electron microscopy images to show the spatial distribution of species; c) X-ray photoelectron spectroscopy (XPS) (probing depth of 10 nm), to investigate with high elemental sensitivity whether silicon is present in natively silicon-free samples of SLBs after incubation with silicones; and d) Infrared spectroscopy (IR), to probe chemical bonds and for the presence of organosilicon compounds.

## Results and discussion

Figure 1a presents the mean Raman spectrum of cells dried in absence (Fig. 1a (1, green line), negative control) and presence (Fig. 1a (2, pink line)) of HMDS. HMDS-dried cells give rise to four main peaks at 490, 710, 2906, and 2964 $cm^{-1}$ that are not detected in the negative control samples. Further confirmation of the presence of HMDS in cells is presented in Figure SI4. The spectrum in Fig. 1a (3, black line) from liquid HMDS shows bands at 569, 685, 2900, and 2958 $cm^{-1}$, which have shifted to 490, 710, 2906, and 2964 $cm^{-1}$, respectively for HMDS in cells. The band positions of HMDS are therefore assigned to an interaction product of HMDS with cellular components, potentially membranes, proteins and sugars, and the formation of silyl ethers. Figure 1b presents Raman images of cells dried with (top row) and without (negative control, bottom row) HMDS, obtained by integration of the band between 450 and 550 $cm^{-1}$, confirming the absence of HMDS-related bands in the negative control cells.

Since AES has a probing depth of only 3 nm, it only detects elements located either *in* or *on* the cell membrane. Therefore, AES was next used to get more insights into the exact localization of the Si species. As depicted on Fig. 1c, in all incubated samples a $Si_{LMM}$ signal was detected, which was almost entirely absent in the negative control samples, while in all samples including the negative controls, as expected, the presence of carbon atoms was revealed.

Using XPS, whose probing depth reaches ~10 nm, brings information across the entire SLB thickness, as well as on the supporting substrate. XPS wide scans (see Fig. 2a) revealed, as expected, the presence of a C 1s signal at 285 eV after formation of a SLB (red), as well as a marked decrease in the signal coming from the ITO (Indium-Tin-Oxide) substrate (e.g., the peak attributed to In 3d at 444 eV). The respective atomic fractions of In 3d, C 1s and Si species per sample indicate that the C 1s signal strongly increased after incubation of the SLBs with any of the organosilicon compounds. This increased carbon signal was accompanied by the emergence of Si 2s and Si 2p peaks at ~153 eV and ~102 eV, respectively, and the concomitant additional decrease of the In 3d signal (Fig. 2b). Noteworthy, a faint Si 2p signal (shifted to ~102.5 eV), having typically an intensity seven times lower than the three categories above, was consistently observed in the non-incubated SLB samples.

The same SLB samples, that were used for AES measurements, were analyzed by IR spectroscopy. Full IR spectra are provided in

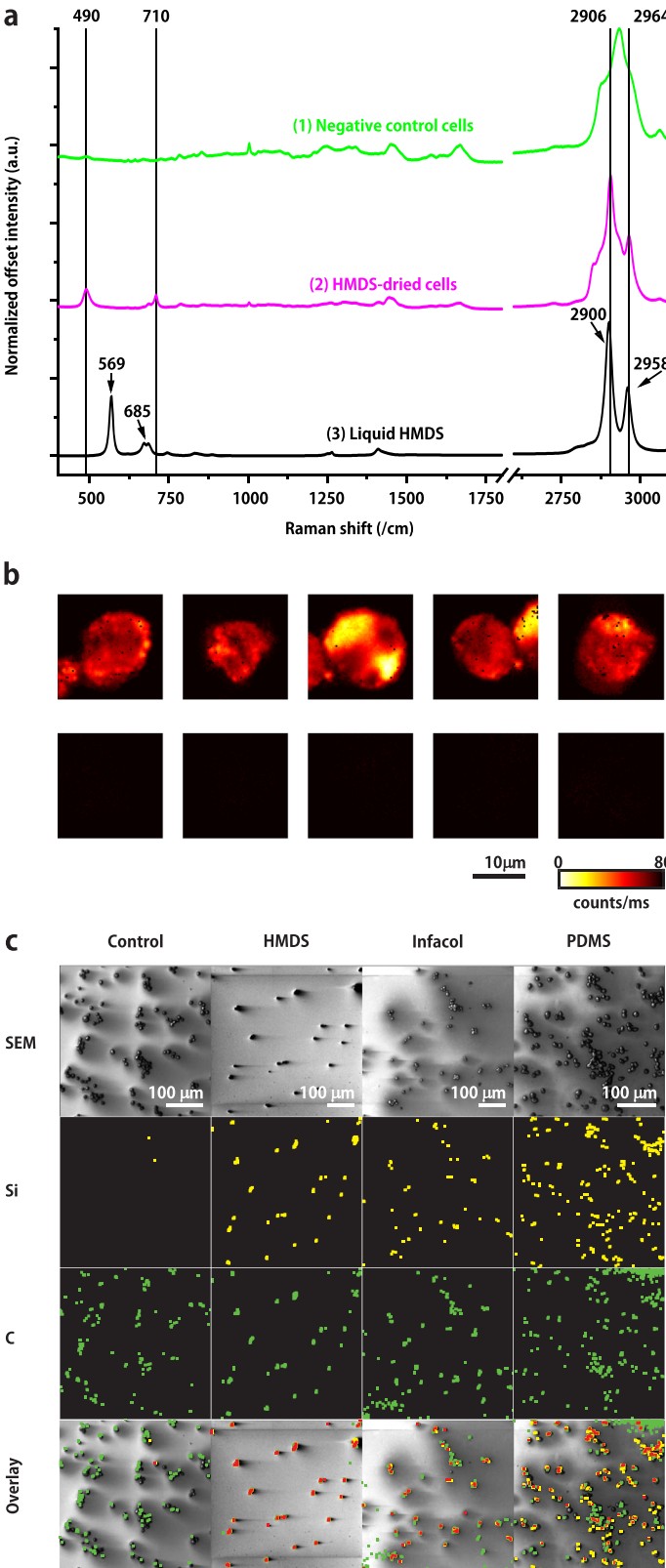

**Fig. 1 Imaging spectroscopic analysis of individual cells. a** Normalized mean Raman spectra of: (1) 5 negative control cells (green), (2) 5 HMDS-dried cells (magenta), (3) neat (liquid) HMDS (black). **b** Raman images of HMDS-dried cells (top row) and negative control cells (bottom row) obtained by integration of the band between 450 and 550 cm$^{-1}$, the region in which the peak assigned to Si–C bonds is located. Raman images were acquired with 35 mW laser excitation power, 100 ms illumination time and 0.31-μm scanning step size. **c** AES/SEM inspection of the silicon (yellow) and carbon (green) content of cells incubated with silicones compared to non-incubated samples (negative controls). In the same locations, overlapping with cells as shown by SEM, both silicon and carbon are found (overlapping dilated pixels corresponding to both Si and C are indicated in red). The original AES spectra revealing also the presence of N 1s in all cells are provided in SI2.

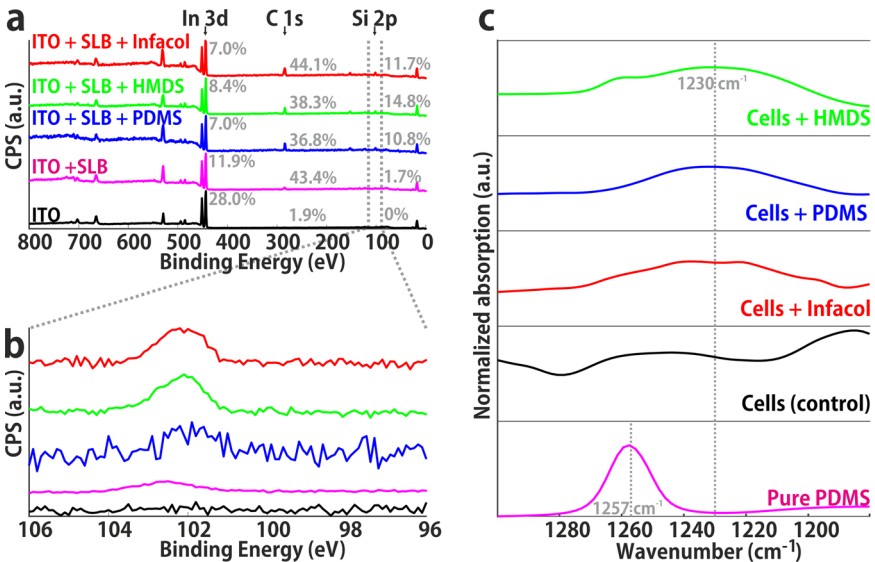

**Fig. 2 Spectroscopic analysis by XPS and IR of biological membranes in the form of SLBs and biological cells. a** XPS wide scan showing all atomic species present; carbon content (285 eV) increases after SLB formation and again after silicone introduction. The relative contribution of In 3d (444 eV) decreases after incubation steps. **b** XPS narrow scan in the Si 2p region, showing increase after silicone introduction. The presence of a trace amount of Si 2p in DOPC supported lipid bilayers (magenta) indicates a minor impurity of the chemicals. **c** Si-C region in IR spectra obtained from pure PDMS (pink) and cells incubated with various organosilicon compounds (green, HMDS; blue, PDMS; red, Infacol; black, negative control). Incubated cells show a shift in a peak (~1250 cm$^{-1}$ → ~1230 cm$^{-1}$) compared to negative control cells.

SI3. Close inspection of the 1180–1300 cm$^{-1}$ region unveils an absorption band for pure PDMS at 1257 cm$^{-1}$, corresponding to the symmetric stretch of the Si-C bond[29]. In the same region, broad bands were found in cell samples incubated with organosilicon compounds, but they were typically red-shifted to ~1230 cm$^{-1}$[27]. Interestingly, this band was absent in all control samples, suggesting that this absorption band could be associated with the presence of organosilicon species in the cell samples. This shift can be the result of a change in environment of the polymer species, e.g., by confinement within the cell membrane[30] and concomitant change in dipolar interactions.

Previously, it was demonstrated elsewhere[31] that small hydrophobic molecules such as drugs and hormones can absorb into the PDMS matrix. Similarly, PDMS can release unpolymerized precursor molecules in solutions, as notably reported by Regehr et al.[6,32]. The results presented here suggest that apolar organosilicon compounds *in general* can embed within various biological *membranes*, driven by physicochemical interactions and not by active uptake.

Raman spectroscopy identified the presence of HMDS in cells dried in its presence. The shift of HMDS-related Raman peaks from liquid, neat HMDS to HMDS-dried in cells suggests that HMDS reacts with molecules in cells and/or their membranes. Raman spectroscopic imaging of single cells (Fig. 1b) reveals that organosilicon compounds are also present intracellularly, at lipid-rich areas, e.g., in the membrane of organelles.

AES results collectively suggest that the presence of Si originates from the incubation of the cells with silicones. Furthermore, these results indicate that these Si-containing compounds are located in the outer 3 nm of the cells, i.e., in or on the plasma membrane of the cells, which does not exclude their presence elsewhere in the cells.

Similar interactions were found in SLBs acting here as simplified models for cell membranes, suggesting that the incorporation of silicones into membranes is a passive process, i.e., not driven by membrane proteins or other endocytic processes. The fact that, using XPS, traces of silicon were observed in DOPC

SLBs, potentially as a result of contamination from ambient organosilicon compounds (e.g., silanes), further illustrates the energetic favorability of apolar organosilicon compounds to interact with the phospholipid aliphatic chains.

The previous results strongly suggest that organosilicon compounds are retained in biological systems and more precisely, associate with lipids in biological membranes. Although the precise interaction is not clear, it is unlikely that a chemical reaction occurred between, on one hand, PDMS and (components of) Infacol and, on the other hand, biological specimens, or that any electrostatic interactions took place, since none of the silicones discussed here are charged. The exact location of the silicones—adsorbed on the outside of the membranes or embedded in the membrane—is at present not clear, as both areas would be observed with AES. Embedding in the membrane is most probable, assuming hydrophobic interactions between the trimethyl silyl moieties and the lipid tail environment.

PDMS and (components of) Infacol were not observed inside the cells, but HMDS, having a much lower molecular weight, was able to transfer into the intracellular compartment. Some understanding can be derived from a thermodynamic argument, which is that the polymer molecules in the vicinity of lipids have lower interfacial energy than those surrounded by water. From this reasoning, it follows that the larger the molecule, the more stable the coordination, which may explain that PDMS oligomers (1.5–6 kDa) and Infacol (14–21 kDa) were not observed inside cells, but HMDS (MW = 161 Da) was able to transfer into the intracellular compartment. To minimize the contact with water, the polymers (with a length of several tens of nanometers) would need to be completely internalized within the phospholipid bilayer (with a thickness of around 5 nm), thereby stretching out to fit in this quasi-2D landscape (Fig. 3, right panel). This reduction in solvation energy[33] is balanced by an entropic cost as it is entropically more favorable for polymers to assume a coiled or globular conformation[34,35] (Fig. 3, middle panel), and since trimethyl silyl groups are large compared to linear alkyl chains. Alternatively, the polymer molecules may also span the membrane in multiple

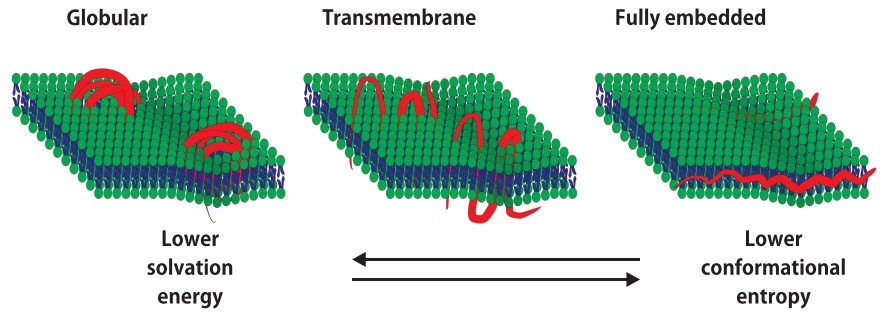

**Fig. 3 Models proposed with different possible conformations for silicone oligomers and polymers in lipid membranes.** Embedding of the polymer within the membrane decreases the solvation energy but is balanced by the entropic cost of uncoiling the oligomer molecule.

regions, much like a transmembrane protein (Fig. 3, middle panel). This compromise would give the organosilicon molecules more fluidity, while overall still resulting in an energetically favorable coordination. To study the actual conformation (which may also depend non-linearly on concentration)[36], computation modeling is required, or the use of advanced optical tools, such as FRAP (Fluorescence recovery after photobleaching), which should reveal changes in the overall lipid bilayer properties. Although the precise conformation of organosilicon compounds in biological membranes is thus not clear, the results presented here demonstrate that interactions occur passively and for multiple types of compounds and membranes.

In conclusion, the results presented here reveal the incorporation of organosilicon compounds in cellular membranes. From this, it can be inferred that the impact of organosilicon compounds on sample preparation, experimental outcome and perhaps even human health should not be ignored. As an example, in studies involving chemical analysis of HMDS-fixed cells[37], it should be noted that the HMDS interferes with the signal. The FDA has cleared several organosilicon compounds for applications in food, cosmetics, and pharmaceuticals on the assumption that these materials are not systemically absorbed[1,38–40]. Knowing that interactions with biological membranes are relatively stable, the notion that silicones are not systemically absorbed needs to be reconsidered.

## Methods

A schematic overview of the sample preparation and subsequent characterization steps is presented in Fig. 4. Each individual step is discussed in more detail in the following sections.

**Materials.** PDMS was prepared using Sylgard 184 (Dow Corning, Farnell, Utrecht, The Netherlands). Infacol (TEVA) was purchased from a local pharmacy store (Die Grenze (Almelo, The Netherlands). 1,2-dioleoyl-sn-glycero-3-phosphocholine was purchased from Avanti Polar Lipids (AL, USA). All other chemicals were obtained from Sigma Aldrich (Zwijndrecht, The Netherlands), unless otherwise specified. In all experiments, phosphate buffered saline (PBS) was prepared at pH 7.4, and filtered through a 0.2-μm syringe filter before use.

Indium tin oxide (ITO) coated fused-silica substrates were used in combination with XPS and Raman spectroscopy. This ITO coating was chosen because of its low Raman background signal. For AES and IR, gold-coated substrates were used. ITO and gold coatings were deposited in the cleanroom of the MESA+ Institute for Nanotechnology by sputtering a layer of ∽100 nm on fused-silica substrates[41]. Thereafter, substrates were cut by dicing to a size of $1 \times 1$ cm$^2$.

**Cell culture.** Cells from a prostate cancer cell line (LNCaP) or colon cancer cell line (HT-29), purchased from the American Type Culture Collection (ATCC, Manassas, VA, USA), were cultured in RPMI-1640 medium with L-glutamine (Lonza, Basel, Switzerland) supplemented with 1% penicillin and 1% streptomycin (Westburg, Amersfoort, The Netherlands) in an incubator at 37 °C and with 5% CO$_2$, medium being refreshed every 3 days and cells being reseeded at a density of $10^4$ cells/cm$^2$. For experiments, $8 \times 10^6$ cells were harvested using a 0.25% Trypsin solution (Thermo Fisher Scientific, Waltham, MA, USA). Subsequently, the cells were fixed in 1% paraformaldehyde (PFA) for 15 min and washed three times in PBS through centrifugation at $300 \times g$ for 5 min. Finally, the sample was split in six

equal fractions, each containing $1.3 \times 10^6$ cells. While fixation alters the chemical state of proteins and other molecules, it was implemented in all experiments for consistency, since it was required for some experiments. It was made sure that the fixative did not contain any organosilicon species not to introduce additional biases in our study.

**Supported lipid bilayers.** Supported lipid bilayers (SLBs) were formed by fusing small unilamellar vesicles on the ITO-coated surface. A DOPC solution in chloroform was dried under vacuum, to yield a lipid film on the walls of a glass vial. This lipid film was re-hydrated in PBS to reach a DOPC concentration of 10 mg/ml and ultrasonicated for 15 min to form small unilamellar vesicles (SUVs). ITO substrates were cleaned ultrasonically in dichloromethane, acetone and ethanol for 3 min each, followed by 30 min of O$_2$-plasma treatment in a Diener Pico (Diener electronic, Bielefeld, Germany) at 250 W. The cleaned surfaces were incubated with a diluted SUV suspension (1 mg/mL in PBS) at room temperature overnight. After incubation, the substrates were thoroughly rinsed with PBS. Before their characterization with XPS, all SLB samples were dried under vacuum overnight.

**Sample preparation—organosilicon.** *Sticky PDMS microchannels* were fabricated using xurography, as previously reported by us[42]. Briefly, Sylgard 184 precursor was thoroughly mixed with the curing agent in a 10:1 weight ratio and degassed by centrifugation at $1000 \times g$ for 1 min. A mold was prepared by cutting a 0.2-mm thick adhesive film to yield $3 \times 6$ mm$^2$ patterns that were laminated in the bottom of a clean petri dish. The PDMS prepolymer/curing agent mixture was poured over the mold and degassed again under vacuum, before being cured at 80 °C for 30 min yielding a sticky solid. Inlet and outlet holes were punched with a Harris Uni-Core 1-mm biopsy punch (VWR International B.V., Amsterdam, The Netherlands). For control experiments, PDMS microchannels were prepared using the same protocol, but more thoroughly cured at 80 °C overnight. Before bonding, the latter PDMS microchannel devices were ultrasonicated in ethanol for 15 min before plasma activation. Solutions were exchanged in these microchannels by pipetting manually in the inlets.

*Infacol* consists of a 40 mg/ml solution of simeticone in water with various additives, e.g., dispersing and flavoring agents. Simeticone consists primarily of polydimethylsiloxane with molecular weight ranging between 14 and 21 kDa, mixed with silicon dioxide nanoparticles (4–7%)[43]. Before use, Infacol was diluted to 1 mg/ml in PBS. Solid particles of few microns in diameter that remained in the solution,were removed by filtering the solution (0.2-μm syringe filter) before experimentation with cells.

*Hexamethyldisilazane* (HMDS) was used as is, from a recently purchased bottle, extracted through a septum under perfusion with nitrogen by a syringe. Prior to HMDS-drying, cells were dehydrated with ethanol. Since Raman bands from ethanol were not observed neither in the HMDS-dried cells, nor in the control cells, it was concluded that ethanol was successfully fully evaporated from the cells.

**PDMS interactions with cells and SLBs.** For AES, cells were alternatively grown on a 1-mm thick PDMS layer prepared in a Petri dish and cured for 30 min at -80 °C. HT-29 cells were seeded at a density of $10^4$ cells/cm$^2$ and left to proliferate for 48 h in RPMI medium supplemented with 1% penicillin and 1% streptomycin at 37 °C in a 5% CO$_2$ atmosphere. Cell adhesion to the PDMS layer after 48 h was comparable to that in standard culture flasks. After trypsinization, cells were fixed for 15 min in 1% paraformaldehyde and washed three times with Milli-Q water by centrifugation ($300 \times g$, 5 min).

Experiments with SLBs were conducted in sticky microchannels placed on the top of cleaned ITO substrates. A DOPC SUV suspension was injected in the microchannel, and left overnight for incubation at room temperature to yield a SLB on the ITO-coated substrate. As for the cells, channels were rinsed with PBS and after delamination of the PDMS microchannels, the substrates were rinsed thoroughly with deionized water before analysis with XPS.

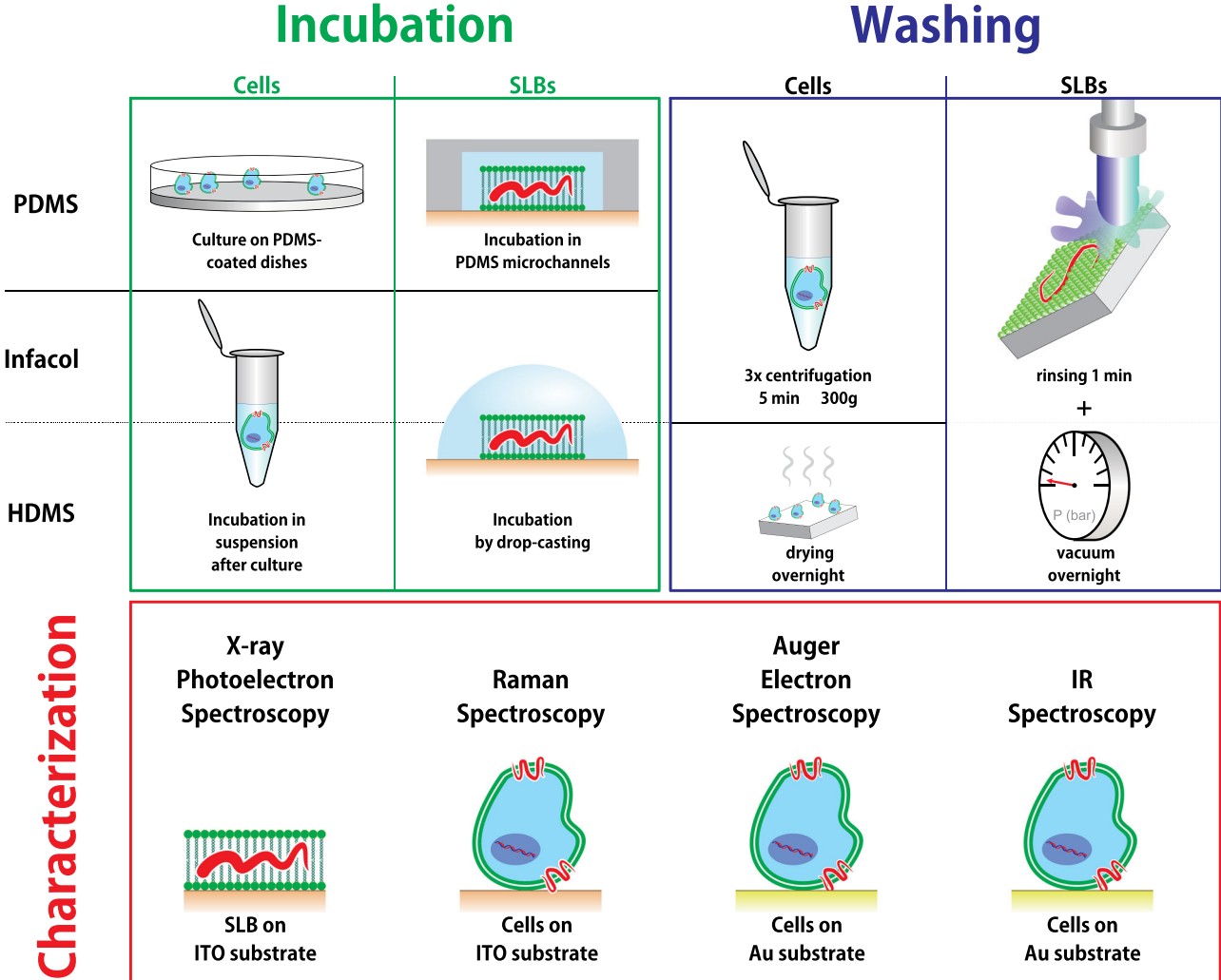

**Fig. 4 Overview of the sample preparation and characterization steps.** Cells and SLBs were first incubated with PDMS, Infacol, or HMDS on substrates conforming to the requirements of the intended characterization technique. The organosilicon compounds are presented as "red wavy lines" and their hypothetical interaction with membranes (vide infra) is here suggested. The excess of organosilicon compounds was removed before samples were characterized with various techniques.

**HMDS interactions with cells and SLBs.** Fixed LnCaP cells in suspension (Mil-liQ) were dehydrated in increasing concentrations of ethanol (70–100%), followed by HMDS-drying[37], and deposited on flat substrates overnight[37]. As a negative control, the HMDS-drying step was omitted and cells in 100% ethanol were dried on the substrates overnight. ITO-coated and gold-coated fused-silica substrates were used for Raman and AES spectroscopy, respectively. SLBs prepared on gold-coated fused-silica substrates were fully immersed in 1 ml of HMDS and dried under vacuum overnight.

**Simeticone interactions with cells and SLBs.** Fixed HT-29 cells were immersed in diluted Infacol solution and left overnight. Before characterization, cells were washed twice in PBS. For AES analysis, cells were deposited on gold-coated silica substrates. SLBs were immersed in 1 ml undiluted Infacol and dried under vacuum overnight after thorough rinsing.

**Raman spectroscopy.** An in-house Raman spectrometer was employed, that has been described in detail elsewhere[42]. Briefly, 2D point scanning of a laser beam ($\lambda = 647.09$ nm) from a Coherent Innova 70C laser was performed. The Raman scattered light was dispersed in a spectrometer and collected with a CCD camera (Andor Newton DU-970-BV, Belfast, UK). The laser power was measured underneath the objective (40×, NA: 0.95; Olympus Nederland B.V., Leiderdorp, The Netherlands,) and adjusted to 35 mW. The laser focal spot was focused 5 μm above the substrate to ensure it was close to the center of the cells. A 20 μm × 20 μm area was scanned with a step size of 0.31 μm and an illumination time of 100 ms per pixel. Hyperspectral images were created by integrating the Raman

band between 450 and 550 cm$^{-1}$, which contains the band at 490 cm$^{-1}$, which is present in all cells dried with HMDS. The area value of each pixel was converted to a color in a heat map scale.

**XPS measurements.** Using XPS, the atomic composition of the SLBs was characterized after incubation with organosilicon compounds (HMDS, PDMS, and Infacol) and compared to control SLBs (no incubation) and bare ITO. Using XPS, with a probing depth of ~10 nm, a signal was detected from the entire SLB and the outer surface of the substrate. As such, this technique gave a comprehensive overview of the elemental composition of a lipid bilayer. Measurements were performed using a JEOL 9200 (JEOL Ltd., Tokyo, Japan) with a monochromatic Al Kα X-ray source operated at 12 kV, with a beam current of 20 mA. The analyzer pass energy was set to 10 eV. Wide scans (0–800 eV) were recorded for inspection of all present elements. Narrow scans in the range of 90–105 eV were acquired to provide more detailed information about silicon presence. Spectra were fitted by Casa XPS software (www.casaxps.com) for quantification.

**Auger electron spectroscopy.** AES was performed using a JEOL JAMP-9500F field emission scanning Auger microprobe (JEOL Ltd., Tokyo, Japan). Briefly, this instrument probes chemical bonds by irradiating a surface locally with a focused electron beam, and measuring the energetic spectrum of electrons emitted through the Auger effect. These secondary electrons with relatively low energy primarily originate from a 2–3 nm layer at the surface. Scanning this beam with a small irradiation spot size allows the acquisition of hyperspectral images with a sub-micron spatial resolution. In conjunction, the instrument can be operated in

scanning electron microscopy (SEM) mode, for comparing the morphological appearance of the sample.

Cells incubated with HMDS and Infacol and cells cultured on PDMS dishes were compared to non-incubated cells as negative controls. Areas with around 20–200 cells were divided into fields of $256 \times 256$ pixels which were scanned in the narrow bands for gold ($Au_{MNN}$, 2015 eV), silicon ($Si_{LMM}$ 92 eV), carbon ($C_{KLL}$ 263 eV) and nitrogen ($N_{KLL}$ 375 eV) with a dwell time of 100 ms per pixel. Narrow band signals were integrated and background subtracted in Spectra Inspection Software (JEOL). The resulting bitmaps were converted to binary images and diluted in ImageJ. Across every row of Fig. 1c, the images were treated with the same threshold settings.

**IR spectroscopy**. The samples inspected with AES were next analyzed using IR spectroscopy on an attenuated total reflection (ATR) with Alpha-P spectrometer from Bruker (Billerica, MA, USA). All spectra were obtained by averaging 32 scans. The resolution was set at 4 cm$^{-1}$. All spectra were recorded at room temperature and ambient atmosphere. These samples were also measured by using reflection FTIR spectra using a 50-µm diameter aperture in a Bruker Hyperion 1000 spectrometer equipped with a 15× objective coupled to a Bruker Tensor 27 FTIR spectrometer. A liquid nitrogen cooled MCT wideband detector was used to detect a spectral range from 4000 to 600 cm$^{-1}$. A background spectrum was collected from plasma-cleaned gold surfaces. In addition to these samples, a gold substrate was homogeneously coated with a thin layer of liquid PDMS to compare the magnitude of the signal from silicone compounds found in cells to those found in pure polymer. Sylgard 184 base and curing agent were mixed in a 10:1 weight ratio and a droplet of this mixture was placed on a cleaned gold surface and spread uniformly using a clean glass microscope slide, resulting in a thin (several µm), coating. This sample was stored at room temperature and measured after 12 h.

**Statistics and reproducibility**. Figure 1a Spectra are derived from ($2 \times 5$) hyperspectral data cubes containing 4096 spectra each, which are shown for a chosen band in Fig. 1b. Figure 1c SEM images of tens of cells in four experiments showing the AES data for Si and C in the same panel. Figure 2 data of a single experiment of five samples for Fig. 2a and Fig. 2b and another five samples for Fig. 2c.

**Reporting summary**. Further information on research design is available in the Nature Research Reporting Summary linked to this article.

## Data availability
The Raman data, concerns about 3GB, but is available to any of our colleagues when asked for. All other data has been uploaded as supplementary data.

## Code availability
The code to prepare Figs. 1a, b and SI4 is written by the authors in Matlab and is available upon reasonable request. The code to prepare Figs. 1c and SI2 is proprietary code that comes with the respective instruments.

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

## Acknowledgements
The authors thank prof. Frans Leermakers for his valuable insights. This work has been conducted as part of the Perspectief Program Cancer ID with project numbers 14193 and 14196, which is funded by the Netherlands Organization for Scientific Research (NWO) and co-financed by JEOL Europe B.V., Hybriscan Technologies B.V.

## Author contributions
P.B., A.E.-M., S.L.G., and C.O. designed the experiment; P.B. and A. E.-M. collected data, P.B. and S.P.P. analyzed AES/SEM, XPS and IR data, A.E.-M. analyzed Raman data; P.B. and A. E.-M. generated figures, P.B., A. E.-M., S.L.G., and C.O. drafted the manuscript, P.B., A.E.-M., S.P.P., L.T. H.Z., S.L.G., and C.O. discussed figures and read and finalized the manuscript.

## Competing interests
The authors declare the following competing interests: C.O. is managing director of Hybriscan Technologies B.V. which partially funded the research. Hybriscan Technologies B.V. has no financial or non-financial competing interest in this study. All remaining authors declare no competing interests.
