## [Transparent Peer Review File · Communications Biology]

Reviewers' comments:

Reviewer #1 (Remarks to the Author):

Organosilicon uptake by biological membranes By Beekman and colleagues addresses a topical and timely issue, but the current presentation of the manuscript fails to deliver a coherent line of argumentation. The high level pitch at the start of the manuscript sets out some big challenges, that the manuscript fails to address. There is a complete lack of discussion of the results and they are not linked back to the key questions posed in the preamble. Overall, while there are some important ideas, the manuscript is not suitable for publication in its present format. I include some recommendations below on how to address the current shortcomings.

The link made in the high level summary between organosilicone and the impact of micro- and nanoplastics is not obvious, and these would definitely not be a significant proportion of microplastic debris? Indeed silicone is currently being touted as an environmentally friendly alternative to plastic. Thus, I would caution about hanging the relevance of the work on the need to understand interactions of microplastics. Maybe consideration of the implications of these findings in terms of silicone as a replacement for plastic would be a more relevant and meaningful framing?

Also in the framing, the last line states that "it is important to understand the bias on 23 characterization results induced by silicone leaching" - however, this is not picked up clearly in the results or discussion.

Lines 67-74: Were all 4 methods applied to both cells and SLBs? A short description of the exposure and sample preparation would be extremely useful to aid in understanding of the next sections- I struggled to follow and had to go and read the methods and then come back and read the rest of the paper.

Lines 177-178: "This would come at an energetic cost, as it is entropically more favorable for polymers to assume a globular conformation (see Figure 3)..." Please check this - my understanding of polymer entropy is that disorder is good - the further the distance between the ends the better (at least for the ideal chain)? A bit more explanation here would be helpful to readers. Figure 3 is not referred to in the text! this figure could be used to further explore the entropy / energy question also.

Lines 89-94: A line on the sample preparation process would be useful here - it is clear for the Raman spectra that volatilisation was the means by which the Si signal should disappear, but in the AES what was the "washing" or clean-up step to remove the "external" source? A schematic figure of the various combinations of methods and cells / SLB and the sample prep methods would be really useful before getting into the interpretation of the results.

Line 284: Not at all clear what is meant by the sentence "for the ITO low Raman background signal and its suitability for chemical surface modification." Ah OK now I get it! it was chosen to these two reasons - please rephrase as it reads like you are assessing the suitability of ITO.

Reviewer #2 (Remarks to the Author):

Summary of the work

The paper describes some basic experiments carried out to study the organosilicon uptake into biological membranes. Authors use imaging and spectroscopy techniques to characterise the interactions of three silica-based reagents with tissue cells. The mechanism of interactions is discussed and possible ways of organosilicon compounds uptake by biological membranes are presented.

General opinion

The study is well designed and described in a clear way. The outcomes of the study could be somewhat expected before conducting it, but the study design allowed to confirm what was already known using unbiased approach, which could be used to study the uptake of other compounds. Therefore, I would recommend this paper to be published after revision, especially of the discussion section where more references and information on the significance of the outcomes should be presented.

Specific comments

In the abstract, please refer to your results mentioning studied compounds.

There is no clear section division in the first part of the paper. Please, define where are the result and discussion sections supposed to start and add the heading for the introduction section.

Lines 67-74 contain a description of experimental design. This should be covered in the Materials and methods section. Hence there is really no need to repeat this in the introduction or results section, unless there is something novel about the approach. Please specify.

Line 83- there is a mention of cells dried without HMDS, please use the description "control" instead. Two sentences between lines 76 and 79 are repetitive. Please remove the first one.

Figure 1 b- please consider removing the second row from the figure 1 b and adding a short description of the result in the text.

Figure 1c- since there are images with overlay, there is no reason really to add the individual elemental maps for Si and C, consider removing

Line 89-94- please rephrase the sentences to make the meaning more clear

Sentence in lines 112-113 should be rather mentioned in the materials and methods section

The discussion part of the paper is well formulated although it could make a use of more referencing.

Lines 297-300- authors mention fixation, please write how this was done or refer to the section where fixation is described

Reviewer #3 (Remarks to the Author):

The study of the potential interaction of organosiloxanes with biological moieties is quite interesting. However, in the current form the study is present in a quite haphazard manner, where experimental details, results, and the inferences drawn are all mixed up. The article needs rewriting with a clear demarcation between the experimental work, the results obtained, and the conclusions drawn from them.

There are several statements in the article that lack supporting evidence - e.g. lines 20-21: '...complexations of these compounds occur in or on the cell membrane and that they can be covalent in nature'. However, rest of the article does not talk about 'complexation' or 'covalent' bonding. If these conclusions are considered valid by the authors, they should explain the nature of such 'complexation' or 'covalent' bonding (how and with what?).

Apart from subjecting to vacuum, there does not seem to be any detail on whether or not the added organosiloxane compounds were removed before processing the cells and model lipid membranes before analytical measurements. If this was not done, what does finding these compounds in dried

samples (albeit with shifted signals) mean? How the authors have excluded the chances of the observed shifts not being artefacts?

Reviewer #1 (Remarks to the Author):

The link made in the high level summary between organosilicone and the impact of micro- and nanoplastics is not obvious, and these would definitely not be a significant proportion of microplastic debris? Indeed silicone is currently being touted as an environmentally friendly alternative to plastic. Thus, I would caution about hanging the relevance of the work on the need to understand interactions of microplastics. Maybe consideration of the implications of these findings in terms of silicone as a replacement for plastic would be a more relevant and meaningful framing?

The authors thank the reviewer for his/her suggestion. We agree that mentioning the broader context may be interpreted as implying a direct link with the results presented in the manuscript and the field of micro- and nanoplastics pollution. The abstract has been adapted accordingly and now reads as:

The increasing application of organosilicon compounds as replacement of other plastics calls for a better awareness and understanding of these interactions.

Also in the framing, the last line states that "it is important to understand the bias on characterization results induced by silicone leaching" - however, this is not picked up clearly in the results or discussion.

The authors thank the reviewer for pointing out this point of improvement. The discussion now reads:

Previously, it was demonstrated elsewhere³⁶ that small hydrophobic molecules such as drugs and hormones can absorb into the PDMS matrix. Similarly, PDMS can release unpolymerized precursor molecules in solutions, as notably reported by. Regehr et al.⁶ The results presented here suggest that apolar organosilicon compounds in general can embed within various biological membranes, driven by physicochemical interactions and not by active uptake.

Furthermore, the following sentence was added:

As an example, in studies involving chemical analysis of HMDS-fixed cells³³, it should be noted that the HMDS interferes with the signal.

Lines 67-74: Were all 4 methods applied to both cells and SLBs? A short description of the exposure and sample preparation would be extremely useful to aid in understanding of the next sections- I struggled to follow and had to go and read the methods and then come back and read the rest of the paper.

The manuscript was originally formatted to have a separate materials and methods section outside of the main text. Adhering to this journal's guidelines, in response to comments of the reviewers the materials and methods section is now embedded in the main text.

Furthermore, Figure 1 was added to clarify these matters.

Lines 177-178: "This would come at an energetic cost, as it is entropically more favorable for polymers to assume 177 a globular conformation (see Figure 3)...." Please check this - my understanding of polymer entropy is that disorder is good - the further the distance between the ends the better (at least for the ideal chain)? A bit more explanation here would be helpful to

readers. **Figure 3 is not referred to in the text! this figure could be used to further explore the entropy / energy question also.**

Figure 3 has been modified . Note that with the addition of Figure 1, the previous Figure 3 is now Figure 4.

A fully extended polymer chain would constitute order: there is only one possible configuration in which that is possible, so there would be a large entropic/energy cost. In absence of external forces, ideal polymer molecules tend to assume a disordered configuration, i.e. a coil. The expected radius is referred to as the Flory radius and can be calculated using statistical arguments. The energy required for elongating a polymer beyond this radius is proportional to the extension of its chain, just like in spring mechanics. Some aspects are now more emphasized in the new text:

To minimize the contact with water, polymers (with a length of several tens of nanometers) would need to be completely internalized in the phospholipid bilayer (with a thickness of around 5 nanometers), thereby stretching out to fit in this quasi-2D landscape (Figure 4, right panel). This reduction in solvation energy is balanced by an entropic cost, as it is entropically more favorable for polymers to assume a coiled or globular conformation (see Figure 3, middle panel) and since trimethyl silyl groups are large compared to linear alkyl chains. Alternatively, the polymer molecules may also span the membrane in multiple regions, much like a transmembrane protein (Figure 4, middle panel). This compromise would give the organosilicon molecules more fluidity, while overall still resulting in an energetically favorable coordination. To study the actual conformation (which may also depend non-linearly on concentration)³⁸, computation modeling is required, or the use of advanced optical tools, such as FRAP (Fluorescence recovery after photobleaching), which should reveal changes in the overall SLB properties.

Lines 89-94: A line on the sample preparation process would be useful here - it is clear for the Raman spectra that volatisation was the means by which the Si signal should disappear, but in the AES what was the "washing" or clean-up step to remove the "external" source? A schematic figure of the various combinatins of methods and cells / SLB and the sample prep methods would be really useful before getting into the interpretation of the results.

We thank the reviewer for his/her excellent suggestion. As mentioned before, Figure 1 was added to clarify the entire experimental process.

Line 284: Not at all clear what is meant by the sentence "for the ITO low Raman background signal and its suitability for chemical surface modification." Ah OK now I get it! it was chosen to these two reasons - please rephrase as it reads like you are assessing the suitability of ITO.

In addition to this remark, the authors noted a mistake in the description of gold surfaces as well.

The text was modified accordingly:

Indium Tin Oxide (ITO)-coated fused-silica substrates were used in combination with XPS and Raman. This ITO coating was chosen because of its low Raman background signal and its suitability for chemical surface modification. For Auger Electron Spectroscopy and IR, gold-coated substrates were used. ITO and gold coatings were applied in the NanoLab cleanroom of the MESA+ Institute for Nanotechnology by sputtering a layer of ~100 nm on fused-silica substrates.²⁸ Thereafter, substrates were cut by dicing to a size of 1×1 cm².

Reviewer #2 (Remarks to the Author):

In the abstract, please refer to your results mentioning studied compounds.

The abstract text was modified and now reads as follows:

Here the interactions of various organosilicon compounds (simeticone, hexamethyldisilazane and polydimethylsiloxane) with cell membranes and model lipid membranes were characterized with a range of analytical techniques demonstrating that these compounds were retained in or on the cell membrane.

There is no clear section division in the first part of the paper. Please, define where are the result and discussion sections supposed to start and add the heading for the introduction section.

Adhering to this journal's guidelines, section headings were added.

Lines 67-74 contain a description of experimental design. This should be covered in the Materials and methods section. Hence there is really no need to repeat this in the introduction or results section, unless there is something novel about the approach. Please specify.

We have adapted the structure of the paper with a *Materials and Methods* section as required by reviewer #1 and reviewer #3, that requested a clearer description of the experiments in the main text

Line 83- there is a mention of cells dried without HMDS, please use the description "control" instead

We added the designation "negative control" as follows:

The same peaks are found in spectra of cells dried with HMDS, yet with a clear shift. Figure 2b shows Raman images of cells dried with (top row) and without (negative control, bottom row) HMDS, obtained by integration of the band between 450 and 550 cm^{-1} .

Two sentences between lines 76 and 79 are repetitive. Please remove the first one.

The authors thank the reviewer for this insight. The first sentence was removed.

Figure 1 b- please consider removing the second row from the figure 1 b and adding a short description of the result in the text.

We thank the reviewer for this suggestion. However, we think that it is important to show the negative controls in the same heat map scale to prove the absence of HMDS in the negative controls.

Figure 1c- since there are images with overlay, there is no reason really to add the individual elemental maps for Si and C, consider removing

The middle rows indeed do not technically add information. However, it was added because it was deemed helpful in interpreting the figure.

Line 89-94- please rephrase the sentences to make the meaning more clear

The text now reads:

Since AES has a probing depth of only 3 nm, it only detects elements located either **in** or **on** the cell membrane. Therefore, AES was next used to get more insights into the exact localization of the Si species. As depicted on Figure 2c, in all incubated samples a Si_{LMM} signal was detected, which was almost entirely absent in the negative control samples, while in all samples including the negative controls, as expected, the presence of carbon atoms was revealed.

Sentence in lines 112-113 should be rather mentioned in the materials and methods section.

The sentence was moved to the *Materials and Methods* section:

Using XPS, the atomic composition of the SLBs was characterized after incubation with organosilicon compounds (HMDS, PDMS and Infacol) and compared to control SLBs (no incubation) and bare ITO. Using XPS, with a probing depth of ~ 10 nm, a signal is detected from the entire SLB and the outer surface of the substrate. As such, it gives a comprehensive overview of the elemental composition of a lipid bilayer.

The discussion part of the paper is well formulated although it could make a use of more referencing.

The authors thank the reviewer for this view and the good suggestion. References 37-40 were added.

Lines 297-300- authors mention fixation, please write how this was done or refer to the section where fixation is described

The fixation was described in line 295. More information has been added in the revised version of the article.

For experiments, 8×10^6 cells were harvested using a 0.25% Trypsin solution (Thermo Fisher Scientific, Waltham, MA, USA). Subsequently, the cells were fixed in 1% paraformaldehyde (PFA) for 15 min and washed three times in PBS through centrifugation at 300 g for 5 min.

Reviewer #3 (Remarks to the Author):

The study of the potential interaction of organosiloxanes with biological moieties is quite interesting. However, in the current form the study is present in a quite haphazard manner, where experimental details, results, and the inferences drawn are all mixed up. The article needs rewriting with a clear demarcation between the experimental work, the results obtained, and the conclusions drawn from them.

The authors value the reviewer's insights and appreciate the opportunity to improve the manuscript. As mentioned before, the Materials and Methods section was embedded in the main text, in response to this comment, among others.

There are several statements in the article that lack supporting evidence - e.g. lines 20-21: '...complexations of these compounds occur in or on the cell membrane and that they can be covalent in nature'. However, rest of the article does not talk about 'complexation' or 'covalent' bonding. If these conclusions are considered valid by the authors, they should explain the nature of such 'complexation' or 'covalent' bonding (how and with what?).

The term "complexation" was chosen to imply that although the bonds between membranes and silicone and PDMS are stable and physicochemical in nature, there is no proof of a covalent bond, nor any reason to suspect any chemical reaction to occur. Since it can also be interpreted as another bonding mechanism, the term "complexation" was removed, e.g. in the following:

are characterized with a range of analytical chemistry tools to demonstrate that these compounds are retained in or on the cell membrane, i.e. that they seem to be systemically adsorbed.

And:

Some understanding can be derived from a thermodynamic argument, which is that the polymer molecules in the vicinity of with lipids have lower interfacial energy than those surrounded by water.

In addition, we have added to the manuscript the following: "The shift in Raman bands between the liquid HMDS and the cells dried with it may suggest a possible reaction of the HMDS with the cell components and the formation of silyl ethers".

Apart from subjecting to vacuum, there does not seem to be any detail on whether or not the added organosiloxane compounds were removed before processing the cells and model lipid membranes before analytical measurements. If this was not done, what does finding these compounds in dried samples (albeit with shifted signals) mean? How the authors have excluded the chances of the observed shifts not being artefacts?

Most notably, all samples were either subjected to washing by centrifugation or rinsing (*i.e.* at intensities much more vigorous than would ever occur *in vivo*).

This was described in the Materials and Methods section, which was given a more prominent position in the main text.

Moreover, it is now graphically illustrated in the newly added Figure 1.

REVIEWERS' COMMENTS:

Reviewer #1 (Remarks to the Author):

new Figure 1 is very nice and helps with the clarity a lot!

Overall, the revised paper is much easier to read / follow and I am happy that it is suitable for publication.

Reviewer #3 (Remarks to the Author):

Thanks for the revision and addressing the comments.

Since the reviewers had no further comments, no changes were made based on their feedback.

REVIEWERS' COMMENTS:

Reviewer #1 (Remarks to the Author):

new Figure 1 is very nice and helps with the clarity a lot!

Overall, the revised paper is much easier to read / follow and I am happy that it is suitable for publication.

Reviewer #3 (Remarks to the Author):

Thanks for the revision and addressing the comments.